# Biomarkers of Favorable vs. Unfavorable Responses in Locally Advanced Rectal Cancer Patients Receiving Neoadjuvant Concurrent Chemoradiotherapy

**DOI:** 10.3390/cells11101611

**Published:** 2022-05-11

**Authors:** Hsin-Hua Lee, Chien-Hung Chen, Yu-Hsiang Huang, Cheng-Han Chiang, Ming-Yii Huang

**Affiliations:** 1Ph.D. Program in Environmental and Occupational Medicine, National Health Research Institutes, Kaohsiung Medical University, Kaohsiung 807, Taiwan; dr.hh.lee@gmail.com; 2Department of Radiation Oncology, Kaohsiung Medical University Hospital, Kaohsiung 807, Taiwan; 1080216@ms.kmuh.org.tw (C.-H.C.); 1060640@ms.kmuh.org.tw (C.-H.C.); 3Department of Radiation Oncology, Faculty of Medicine, College of Medicine, Kaohsiung Medical University, Kaohsiung 807, Taiwan; 4Center for Cancer Research, Kaohsiung Medical University, Kaohsiung 807, Taiwan; 5Department of Radiation Oncology, Kaohsiung Municipal Ta-Tung Hospital, Kaohsiung Medical University, Kaohsiung 801, Taiwan; 6Post-Graduate Year Training, Kaohsiung Medical University Hospital, Kaohsiung 807, Taiwan; johnhuang851205@gmail.com; 7Graduate Institute of Medicine, College of Medicine, Kaohsiung Medical University, Kaohsiung 807, Taiwan

**Keywords:** CCRT, rectal cancer, biomarker, pCR, neoadjuvant therapy

## Abstract

Colorectal cancer is the second leading cause of cancer death globally. The gold standard for locally advanced rectal cancer (LARC) nowadays is preoperative concurrent chemoradiation (CCRT). Approximately three quarters of LARC patients do not achieve pathological complete response and hence suffer from relapse, metastases and inevitable death. The exploration of trustworthy and timely biomarkers for CCRT response is urgently called for. This review focused upon a broad spectrum of biomarkers, including circulating tumor cells, DNA, RNA, oncogenes, tumor suppressor genes, epigenetics, impaired DNA mismatch repair, patient-derived xenografts, in vitro tumor organoids, immunity and microbiomes. Utilizing proper biomarkers can assist in categorizing appropriate patients by the most efficient treatment modality with the best outcome and accompanied by minimal side effects. The purpose of this review is to inspect and analyze accessible data in order to fully realize the promise of precision oncology for rectal cancer patients.

## 1. Introduction

Colorectal cancer (CRC) is the second leading cause of cancer death (9.4%) and the third most commonly diagnosed cancer (10.0%) in the latest global cancer statistics using the GLOBOCAN 2020 estimates of cancer incidence and mortality by the International Agency for Research on Cancer [1]. Neoadjuvant Concurrent chemoradiation (CCRT) before surgical intervention is an imperative component in the current standard treatment for locally advanced rectal cancer (LARC) [2,3,4,5]. However, the challenge that a wide variety of treatment responses occurs after similar standardized management remains [6,7]. Approximately 15–27% of patients achieve complete pathological response (pCR—ypT0N0), whereas about three-quarters of patients do not and thus have higher rates of relapse and death [8]. A mounting body of evidence strongly supports the paramount importance of biomarkers to provide individually-tailored therapeutic recommendations [9,10,11]; however, until recently, all patients requiring CCRT have had to balance the benefits and potential risks of toxicity due to lack of established and reliable predictive markers in clinical use.

Some studies have highlighted various biomarkers that could be considered in reaching the best treatment decisions for each patient on an individual basis [12,13,14,15], although evidence-based research on the role of biomarkers in predicting response to neoadjuvant CCRT in patients with LARC remains scarce. The European Society for Medical Oncology (ESMO) consensus guidelines consider downgrading in Tumor or Nodal classification as a prognostic factor of favorable outcome rather than a predictive biomarker for further adjuvant treatment [16]. Controversial viewpoints enable further clarification, and the purpose of this review is to scrutinize the most updated data and analyze not only novel biomarkers but comprehensive and timely methods in clinical practice as well. This ought to help minimize the potential barriers and fully realize the promise of precision oncology for cancer patients. In this review, we will specifically summarize the convenience and efficacy of available biomarkers regarding tumor response after neoadjuvant CCRT and not survival, which is multifactorially influenced.

## 2. Circulating Tumor Cells

In 1955, Engell published a clinical study on the occurrence of cancer cells in the peripheral blood and in venous blood draining of the tumor area during operation [17]; following this, the presence of circulating tumor cells (CTCs) was then documented [18]. Moreover, real-time quantitative polymerase chain reaction (PCR) and high-sensitivity colorimetric membrane array-based multigene biomarker assays can detect CTCs in the peripheral blood [19,20,21]. Studies a decade ago showed MicroRNA (miRNA) is significantly elevated in the plasma of patients with CRC and can be a potential marker for screening. MiR-17-3p and miR-92 were significantly elevated in the patients, and the plasma levels of these miRNAs were reduced after surgery [22,23].

Sun et al. reported the advantage of detecting CTC level via EpCAM magnetic bead-based enrichment combined with cytometric identification over serum carcinoembryonic antigen (CEA) for predicting treatment responses in rectal cancer [24], finding a close correlation between CTC levels and treatment outcomes [24]. In addition, they measured the concentration, KRAS mutation and O6-methylguanine-DNA methyltransferase (MGMT) promoter methylation status of cell-free DNA by PCR [25]. The 400-base pair (bp) DNA concentration and 400-/100-bp DNA ratio dropped notably after CCRT in the good responders, and higher MGMT promoter methylation status at baseline DNA was associated with better tumor response [25].

Tumor regression grading (TRG) was defined as follows: grade 0, no regression; grade 1, dominant tumor mass with obvious fibrosis and/or vasculopathy; grade 2, dominantly fibrotic changes with few tumor cells or groups (easy to find); grade 3, very few tumor cells (difficult to find microscopically) in fibrotic tissue with or without mucous substance; grade 4, no tumor cells, only fibrotic mass (total regression or response). Baseline CTC counts of responders with tumor regression grade (TRG) classifications 3–4 were appreciably superior to those of non-responders (TRG0-2) (44.50 ± 11.94 vs. 37.67 ± 15.45, *p* = 0.012) [26], and the post-CCRT CTC counts of responders were extensively lower than those of non-responders (3.61 ± 2.90 vs. 12.08 ± 7.40, *p* < 0.001). The results of multivariate analyses indicated that post-CRT CTC counts and ∆%CTC (percentage difference in CTC counts between baseline and post-CCRT) were significantly and independently related to CCRT response [26].

Furthermore, in a prospective single institutional study by Magni et al., there was a statistically noteworthy association between changes of CTC number from before to after neoadjuvant chemotherapy, according to pathological responses in their cohort of cT3–4 and/or N+ rectal cancer patients treated with neoadjuvant CCRT (*p* = 0.02) [27]. To summarize, the predictability of CTCs might require more (and larger) prospective research studies in order to authenticate its credibility.

## 3. DNA and RNA

However, Tie et al. reported that the conversion of circulating tumor DNA (ctDNA) status from positive at baseline to negative at four to six weeks after CCRT was not correlated with pCR (pCR vs. non-pCR, 95% vs. 88%, *p* = 0.46) in a prospective study [8], although the latest study involving 29 patients with LARC confirmed that ctDNA predicts pCR. The overall margin-negative, node-negative resection rate was 73% and was considerably higher among patients with undetectable preoperative ctDNA (*n* = 17, 88%) versus patients with measurable preoperative ctDNA (*n* = 9, 44%; *p* = 0.028) [28]. Table 1 summarized five LARC studies of CTCs and ctDNA characteristics regarding tumor response after neoadjuvant treatment.

The gene interaction network and module analysis of differential expression mRNAs contained in the lncRNA-miRNA-mRNA network identified five hub genes (KRAS, PDPK1, PPP2R5C, PPP2R1B and YES1) closely associated with CCRT response in LARC [29]. Three lncRNA-based signatures: lnc-KLF7-1, lnc-MAB21L2-1 and LINC00324 were found to be the most promising variable subset for classification, with overall sensitivity and specificity of 0.91 and 0.94, respectively, and with an AUC of our ROC curve = 0.93 [30]. Palma et al. used the Human WG CodeLink microarray platform and demonstrated that high Gng4, c-Myc, Pola1 and Rrm1 mRNA expression levels were significant predictors for CCRT response in LARC patients (*p* < 0.05) [31]. Aberrant DNA methylation, specifically the cytosine-phosphate-guanine (CpG) island methylator phenotype (CIMP), has been displayed in CRC [32]. Some researchers proposed a novel prophetic tool based on three CpGs potentially useful for pretreatment screening of LARC patients and steering the selection of treatment modality [33]. Lately, MicroRNA-130a (miR-130a) has been spectacularly upregulated in radiosensitive rectal cancer cells where overexpression of miR-130a promotes rectal cancer cell radiosensitivity by targeting SOX4 [34].

## 4. Oncogenes and Tumor Suppressors

The pursuit of biomarkers to differentiate responsive patients is to avoid drastic procedures causing inadvertent iatrogenic complications. Genomic data pertaining to oncogenes and tumor suppressors have long been considered potential gizmos to select better-responsive patients for less radical strategies. Modern technology enables us to profile over five hundred genes to deliver a comprehensive genomic profile at a time. With access to cancer profiles that account for the unique molecular profile and biology of rectal cancer in an individual person, researchers have aimed at finding the optimal predictive biomarkers. The drug 5-FU is catabolized in vivo via the dihydrothymine dehydrogenase (DPYD/DPD) [35]. Huang et al. tested multiple genetic biomarkers (chip), including DPYD, TYMS, TYMP, TK1 and TK2, at a cutoff value for three positive genes, and a sensitivity of 89.7% and a specificity of 81% were obtained (AUC: 0.915; 95% CI: 0.840–0.991) [36]. Negative data on chips significantly correlated to poor neoadjuvant CCRT responses (TRG 0–1) (*p* = 0.014, hazard ratio: 22.704, 95% CI: 3.055–235.448 in multivariate analysis) [36].

Mutation of TP53 was most frequently detected in stage III patients with tumorigenicity [37]. More than two decades ago, Adell et al. proposed p53 as an indicator for the effect of preoperative radiotherapy (RT) of rectal cancer using data from the Southeast Swedish Health Care region included in the Swedish rectal cancer trial between 1987–1990 [38]. Later, Kim et al. discovered that immunohistochemical expressions of p53 and BCL-2 had no correlation with tumor response after CCRT, but Ki-67 labeling might be a useful parameter for radiosensitivity [39]. In light of the inconsistent results regarding p53 status and the response to neoadjuvant CCRT, a meta-analysis that included 1830 cases documented the wild-type form of p53 status (low expression of p53 protein and/or wild-type p53 gene) being associated with pathologic responses in rectal cancer patients who received neoadjuvant radiation-based therapy (good response: risk ratio [RR] = 1.30; 95% confidence intervals [CI] = 1.14–1.49; *p* < 0.001; complete response RR= 1.65; 95% CI = 1.19–2.30; *p* = 0.003; poor response RR= 0.85; 95% CI = 0.75–0.96; *p* = 0.007) [40].

Tumors with TP53 mutations tend to accumulate through CCRT [41]. A pilot study by Douglas et al. discerned that patients with partial response (PR) showed mutations in four genes that were not mutated in complete responders (CR): KDM6A, ABL1, DAXXZBTB22 and KRAS [42]. Ten genes were mutated only in the CR group, including ARID1A, PMS2, JAK1, CREBBP, MTOR, RB1, PRKAR1A, FBXW7, ATM C11orf65 and KMT2D, with specific discriminating variants noted in DMNT3A, KDM6A, MTOR, APC and TP53 [42].

The mammalian Rat Sarcoma Viral Oncogene Homolog (Ras) gene family consists of the Harvey, Kirsten Ras genes and the Neuroblastoma RAS Viral Oncogene Homolog (NRas) gene. HRAS, KRAS and NRAS are the most common oncogenes in human cancer [43]. The KRAS gene is the most frequently mutated (about 22% in all human tumors) among the three isoforms, followed by NRAS (8%) and HRAS (3%) [44]. KRAS mutations have been detected in 35–45% of all patients with CRC [45,46,47], whereas NRAS and HRAS were only found in less than 5%. A retrospective observational study of Japanese patients with metastatic CRC found that among 264 patients, mutations in KRAS exon 2, KRAS exons 3 or 4 and NRAS were detected in 34.1%, 3.8% and 4.2% of patients, respectively [48]. Another American study performed molecular testing on 1286 consecutive metastatic CRC from 1271 patients as part of routine clinical care, and RAS amplification was detected in and included: KRAS, NRAS and HRAS for 15, 5 and 2 cases, respectively [49]. A high neutrophils-platelets score (NPS) (OR = 10.52; 95% CI = 1.34–82.6; *p* = 0.025) and KRAS mutated cases (OR = 5.49; 95% CI = 1.06–28.4; *p* = 0.042) were identified as independent predictive factors of poor response to neoadjuvant CCRT in a multivariate analysis with non-metastatic rectal cancer [50].

In addition to TP53, Luna-Pérez et al. reported that specific KRAS mutations are an indicator of tumor response in patients with LARC treated with preoperative CCRT and surgery [51]. KRAS is a protein downstream of the epidermal growth factor receptor (EGFR; also known as HER1). The rate of KRAS mutation was reduced considerably after CCRT, whether the patients had a good or poor response [25]. Another study showed that KRAS mutation was independently related to a lower pCR rate in patients with LARC after adjusting for variations in the neoadjuvant regimen [52].

Some authors had different results. Davies et al. reported that Phospho-ERK and AKT status, but not KRAS mutation status, were associated with clinical outcomes in LARC treated with CCRT [53]. PI3K/AKT/mTOR is an intracellular-signaling pathway regulating cell cycles. There is ample evidence regarding PI3K/AKT/mTOR pathway inhibitors as improving radiotherapy response in rectal cancer but not so much in clinical practice at present [54]. Chow et al. considered that KRAS and combined KRAS/TP53 mutations in LARC were independently associated with poor pCR after neoadjuvant CCRT in a retrospective analysis of 229 pretreatment biopsies [52]. The early detection of CTCs with KRAS mutation might guide therapy in certain patients [55].

Some studies have revealed many discriminating genes that could be used for depiction of CCRT response [42,56,57,58], and more recently, the highest accuracy rate (89.1%) has been displayed by using four genes: LRRIQ3, FRMD3, SAMD5 and TMC7, with a predictive accuracy rate of 81.3% [59]. DNAJC12 is a member of the HSP40/DNAJ family that was revealed to be a potential biomarker in a study conducted with 172 patients as high expression of DNAJC12 was notably associated with inferior TRG (*p* = 0.009) [60]. The SMAD3 gene is involved in the cytoskeleton remodeling pathway [61]. A multivariate analysis showed that phosphorylated SMAD3 overexpression correlated to poor neoadjuvant CCRT responses (*p* = 0.015; OR 7.218; 95% CI 1.479–35.229), and pathological TRG of 0–1 was an independent predictor of postoperative relapse (*p* = 0.021; OR 5.452; 95% CI 1.286–23.113) [61].

Excision repair cross-complementing (ERCC) genes encode proteins to remove DNA lesions and maintain chromosome stability [62]. In this complex DNA repair mechanism, ERCC overexpression was associated with the poor response of neoadjuvant CCRT in rectal cancer [62,63]. Apoptosis inducers (lumican, thrombospondin 2, and galectin-1) have been proven to have higher expression in responders, whereas apoptosis inhibitors (cyclophilin 40 and glutathione peroxidase) have higher expression in non-responders after CCRT [56]. Another study identified 95 genes displaying differential expression between PR and CR [64]. He et al. determined that high expression of Regenerating Gene Type IV (REG4) was the most significantly associated gene with CCRT resistance [65]. Table 2 summarizes the above studies.

## 5. Epigenetics-Gene Methylation Transcriptome/Epigenome

Various types of epigenetic processes have been identified: methylation, acetylation, phosphorylation, ubiquitylation and sumoylation. In 1983, Feinberg and Vogelstein originally found that CRC had less DNA methylation than normal tissue from the same patients [66]. Analysis of the association between methylation and response to therapy in tumor samples showed that only TIMP3 gene methylation status differed significantly within the four TRG classes (ANOVA, *p* < 0.05) [67].

In terms of the fibroblast growth factor/fibroblast growth factor receptor (FGFR) signaling pathway, scientists have also performed experiments to better excel in grasping its core function. The mammalian fibroblast growth factor (FGF) exerts its actions through four vastly conserved transmembrane tyrosine kinase receptors, including FGFR1, FGFR2, FGFR3 and FGFR4. This signaling pathway controls cellular processes in different frameworks, including proliferation, differentiation, survival and motility. High expression of FGFR2 appears significantly linked to inferior TRG (*p* < 0.001) [68].

## 6. Impaired DNA Mismatch Repair

The exquisite DNA in our bodies may encounter impaired DNA mismatch repair (MMR) where such deficiency is a replication error status that can result in microsatellite instability (MSI). The 5-fluorouracil response is strongly affected by MMR deficiency and the loss of heterozygosity for *DCC* (chromosome 18) and mutations in TGFbIIR are secondary [69]. Because current CCRT is 5-fluorouracil-based, MMR ought to be taken into account. Fluoropyrimidine-based neoadjuvant CCRT is associated with a pCR rate of 27.6% in 62 patients with DNA MMR deficiency [70]. Charara et al. reported that high MSI with a loss of MMR protein expression and p21WAF1/C1PI is predictive of superior CCRT response in LARC patients [71], and in their study, tumors with CR showed higher expression of bcl-2, Ki-67, topo II and p27. Furthermore, p53 is more frequently expressed in PR tumors [71].

A recent study used immunohistochemistry, laser capture microdissection/qRT-PCR, flow cytometry and functional analysis of tumor-infiltrating lymphocytes (TIL) from CRC patients and unveiled that the dynamic immune microenvironment of MSI was balanced by multiple counter-inhibitory checkpoints, such as programmed cell death 1 (PD-1), PD-L1, cytotoxic T cell-associated protein 4 (CTLA-4), LAG-3 and IDO [72]. There have been good outcomes in studies with an immune checkpoint blockade by PD-1 receptor inhibitors, nivolumab and pembrolizumab and the CTLA-4 inhibitor, ipilimuma [73,74]. Avelumab is a fully human immunoglobulin that binds PD-L1. A phase II trial found that in patients with LARC, neoadjuvant radiation followed by mFOLFOX6 with avelumab is safe, with a promising pCR [75].

Focusing on some genes as regulators of radio-sensitivity, the study on MSI continues. The overexpression of CHD4 resulted in radio-resistance in MSI-High (MSI-H) colorectal cells, whereas the knockdown of CHD4 enhanced radio-sensitivity in Microsatellite stable (MSS) cells [76]. Wang et al. utilized the microarray datasets (GSE68204) of rectal cancer from the Gene Expression Omnibus database and confirmed overexpression of CHD4 is an independent biomarker of poor TRG and worse therapeutic response in LARC patients after CCRT [76]. Rectal tumors with deficiency of MMR are sensitive to CCRT [77], and even though MSI is independently linked to less pCR for LARC patients with 4450 MSS and 636 MSI in a National Cancer Database (NCDB)-based analysis [78], some researchers might hence deem MMR status as a prospective biomarker for CCRT response.

However, the majority of our rectal cancer patients are MSS with still existing diverse CCRT responses. Biomarkers may play a more preponderant role if they can be applied to most populations. B-cell-specific Moloney murine leukemia virus insertion site 1 (BMI1) deficiency enhances radiosensitivity in MSS CRCs, whereas BMI1 overexpression was found to significantly correlate with inferior TRG (*p* = 0.001) in 172 patients with LARC [79]. In 2020, a meta-analysis of five studies describing 5877 patients found no significant difference in pCR rate following CCRT in patients with MSI versus MSS rectal tumors [80]. There may still be hitherto undiscovered mechanisms of MMR to CCRT response.

For a minority of patients with MSI-high/MMR-deficient tumors, either due to Lynch syndrome or sporadic mutations, immunotherapy is recommended as first-line treatment [81]. There have been several case series of MMR-deficient LARC that showed significant response with neoadjuvant immunotherapy-based systemic treatment [82,83,84]. Immune checkpoint inhibitors are more active in treatment-naïve patients than in those with refractory MSI-H/deficiency in MMR CRC [11,73,84]. Since neoadjuvant CCRT modulates the immune-related characteristics of LARC, it may thus enhance the responsiveness of LARC to immunotherapy [85,86].

Histones are proteins abundant in lysine and arginine residues that are found in eukaryotic cell nuclei. In another study, high histone ubiquitination enzyme (UBE2B) expression appeared significantly correlated with poor TRG, and the recruitment of 53BP1 and Rad51 was remarkably prolonged in cells after pre-treatment with the UBE2B inhibitor TZ9, suggesting a defective DNA repair pathway in UBE2B-deficient cells [87]. Figure 1 displays the crucial parameters for favorable responses.

## 7. Patient-Derived Xenograft (PDX) and In Vitro Tumor Organoid (PDO)

Successful disease-specific preclinical models are still absent. Guenot et al. reported in 2006 that patient-derived xenografts (PDXs) retain primary tumor genetic alteration and intra-tumoral heterogeneity and remain stable across passages [88]. PDX models are progressively exploited in translational cancer research [37,89]. Organoid technology has recently been recommended for generating an imitation to reflect the value in ex vivo drug testing [90]. Nunes et al. investigated 52 colorectal PDXs, composed of 48 MSS and 4 MSI tumors and reached appealing findings regarding IGF2–PI3K and ERBB–RAS alterations in colorectal tumor grafts. The genomic anomaly frequencies observed in MSS PDX reproduced those detected in non-hypermutated patient tumors. There was frequent IGF2 upregulation (16%), which was mutually exclusive with IRS2, PIK3CA, PTEN and INPP4B alterations, supporting IGF2 as a prospective drug target [91]. On the other hand, since G9a is crucial in mediating phenotypes of cancer stem cells (CSCs), a study with PDX in immunodeficient mice and in vitro stemness ability showed evidence of surviving cells after RT with high levels of G9a [92]. They discovered a positive correlation between G9a and the CSCs marker CD133 in LARC patients with CCRT, and that knockdown of G9a increased radiosensitivity and sensitized cells to DNA-damage agents through the PP2A-RPA axis [92].

Organoids and PDX models have similar clonal selections and heterogeneity [93]. Patient-derived organoids (PDOs) are unlike in vivo PDX in that the former preserves the tumor microenvironment (TME) architecture with tumor parenchyma and stroma, including tumor-specific TILs [94,95]. Some investigators have established a biorepository of PDOs to mimic the genetic multiplicity and CCRT response of rectal cancer [96,97]. The neoadjuvant CCRT responses in PDOs in vitro were replicated as observed clinically in corresponding patient tumors (*p* < 0.05) [98]; additionally, CCRT responses in patients corresponded to PDO responses with 84.43% accuracy, 78.01% sensitivity and 91.97% specificity [99]. Nonetheless, the present direction of PDO studies lies in reproducibility and accuracy, as the similarity of these models to patient tumors could be as low as only 40% [100]. Because PDOs are more difficult to create from those with MSI, if they areBRAF-mutated, poorly-differentiated and/or of a mucinous type, they could barely be applied in patients with the aforementioned tumor characteristics [90]. The combined targeting of EGFR and KRAS (G12C) is highly effective in CRC cells, PDO and PDX, suggesting a new therapeutic strategy to treat patients with KRAS (G12C)-mutated CRC [101]. Since PDO-based in vitro cell culture models could preserve the histological and mutational characteristics of their corresponding tumors and mimic the tumor microenvironment, PDX or PDO has been employed for drug screening in the last decade.

## 8. Immunity

CCRT boosts local immune response by increased TILs. The density of CD8+ TILs in post-CCRT resected specimens was significantly increased compared with that in pre-CCRT biopsy samples [102]. In addition, radiotherapy fractionation significantly influenced the CD8C/FoxP3C ratio after CCRT (*p* = 0.027), with a lower ratio with hypofractionated RT [103]. More than a decade ago, Yasuda et al. examined the numbers and the densities of both CD4 + and CD8+ TIL in pre-CCRT biopsy samples and found that they were strongly correlated with the tumor reduction ratio and histological grade, respectively. The density of CD8+ TIL is an independent factor for CR [104], and alterations in the densities of TILs have continued to draw the attention of scientists. Teng et al. reported high pretreatment CD3+ and CD8+ TILs were associated with a good response (TRG ≥ 3) (*p* = 0.033 and 0.021, respectively) [105]. CD8C/GrzBC T-cells in the tumor stroma are significantly associated with poor TRG [106]. Because the proportions of patients with high densities of CD3+, CD4+, CD8+ and FoxP3+ cells seven days after starting CCRT were radically lower than the respective values before starting nCRT (*p* < 0.0001, *p* < 0.0001, *p* = 0.0023, and *p* = 0.0046), some authors have recommended the evaluation of immunohistochemical staining ought to be conducted after CCRT and not prior to CCRT [107]. Figure 2 displays the crucial parameters for unfavorable responses.

Whole-exome sequencing and gene expression microarray analysis were conducted to investigate the genomic properties associated with CCRT response, with a study in 275 patients showing that pre-CCRT CD8+ TIL density was associated with better CCRT response [108]. Whole-exome sequencing in 74 patients showed that the numbers of single-nucleotide variations (SNVs) and neoantigens predicted from SNVs were higher in good responders than in their counterparts, and these correlated positively with CD8+ TIL density (rS = 0.315 and rS = 0.334 respectively) [108]. Ample evidence has shown that T-cell complexity and density are associated with sensitivity to neoadjuvant CCRT, where several biomarkers have been elected, such as Tumor-infiltrating FOXP3+ T regulatory cells and the density of total and cytotoxic T lymphocytes, CD8+ TIL [109,110,111,112,113,114,115,116].

Systemic inflammation has long reflected on clinical outcomes of many diseases and a systemic inflammatory response before CCRT was proven to be linked to poor pCR [117,118]. Elevated preoperative Neutrophil-to-lymphocyte ratio (NLR) is caused by neutrophilia and/or lymphopenia, the two conditions of the pro-tumor inflammatory process. There are tight connections between systemic inflammation and nutritional status in the host. NLR has been evaluated with cutoffs before and after CCRT, as 2.8 and 3.8 respectively, whereas preoperative NLR higher than cutoffs had significantly poor TRG and postoperative complications [119]. A recent retrospective study was conducted on 1052 patients operated on during 2013–2019 and calculated that an NLR value of ≥3.11 indicated poor responses to neoadjuvant CCRT [120].

A study with 170 patients validated that post-CCRT CEA levels < 5 ng/mL were associated with increased rates of clinical CT and pCR [121]. CEA alone failed to significantly impact pCR in another study of 562 LARC patients in a multivariate analysis [122]. A preoperative Fibrinogen–Albumin ratio index (FARI) was deemed to be a reliable CCRT response predictor (OR = 3.044, *p* = 0.012) in LARC Patients [123]. A high percentage of FARI patients (>8.8%) showed poor CCRT response. The same research team built a scoring system named CEA-FARI-PNI (CFP) that combined CEA, FARI and the Prognostic Nutritional Index (PNI) and found that high CFP (OR  =  3.693, *p*  =  0.002) was an independent risk factor for poor response [124]. The combination score enhanced the predictive value than any one of the three factors alone. Wang et al. proposed the optimal cutoff value of the PNI was approximately 45 [125].

## 9. Microbiome

Gut microbiome analysis is a recently rising yet imperative field of study. The gut microbiota harbor about 100 trillion microbial cells, constituting a complex community of bacteria, fungi, protozoa and viruses [126]. The level of gram-positive bacteria potentially influences radiotherapy response [127], as radiotherapy has immune modulatory effects via tumor-associated antigen cross-presentation to cytolytic CD8+ T cells and IFN-γ. Toomey et al. suggested that tumor Fusobacteria may be associated with poor TRG following CCRT in LARC [128]. A study of 1041 patients with CRC revealed the association of F. nucleatum with TIL differed by tumor MSI status [129]. Moreover, Bacteroidales (Bacteroidaceae, Rikenellaceae, Bacteroides) were relatively more copious in patients with non-CR than those with CR and Duodenibacillus massiliensis, and was linked with a better CR rate, according to a Bayesian network analysis [130]. Microbes related to butyrate production including Roseburia, Dorea and Anaerostipes were overrepresented in responders, whereas Coriobacteriaceae and Fusobacterium were overrepresented in nonresponders [131]. In mouse models, buccal Fusobacterium nucleatum migrated rectal cancer lesions and decreased the therapeutic efficacy and prognosis of radiotherapy [132].

Yi et al. found that Dorea, Anaerostipes and Streptococcus yielded an area under the curve value of 93.57% [95% CI, 85.76–100%] in the training cohort and 73.53% (95% CI, 58.96–88.11%) in the validation cohort [131]. Figure 3 illustrates different species of microbiota related to favorable and unfavorable neoadjuvant CCRT responses in LARC patients, respectively.

## 10. Conclusions

Predictive biomarkers for neoadjuvant CCRT response in LARC patients have developed within reach. CTC and genetic profiling including epigenetic exploration once proposed has now been customized, with new technology whose credibility and feasibility continues to improve. Meanwhile, many genes have been discovered to have a correlation with neoadjuvant CCRT responses. Certain pathway investigations in the FGFR signaling pathway, DNA repair and PI3K/AKT/mTOR pathway have been further clarified under investigation and some inhibitors may soon enter clinical trials.

Radiosensitization relies on the alteration of the tumor microenvironment with integration of proper immunity and nutrition of the distinct host. The implementation of reliable biomarkers for treatment stratification and clinical management requires validation in large independent studies, which are now warranted. Tumor grafts from either xenografts or tumor organoids face the technical challenges of accuracy, sensitivity and specificity. Some novel biomarkers, such as gut microbiomes, entail prospective validation to guide clinicians in optimal decision making. In the prediction of CCRT response prior to surgical intervention for LARC patients, there have been recent advancements, leading to an expandingly wide selection.

## Figures and Tables

**Figure 1 cells-11-01611-f001:**
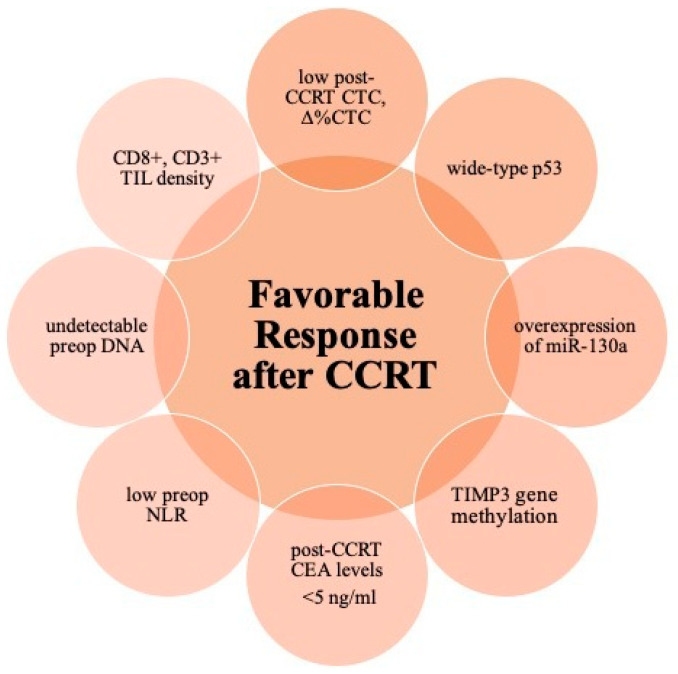
Schematic diagram of favorable biomarkers for neoadjuvant concurrent chemoradiation in patients with locally advanced rectal cancer. Abbreviations: CCRT, concurrent chemoradiation; CTC, circulating tumor cells; CEA, carcinoembryonic antigen; NLR, Neutrophil-to-lymphocyte ratio; preop, preoperative; TIL, tumor-infiltrating lymphocytes.

**Figure 2 cells-11-01611-f002:**
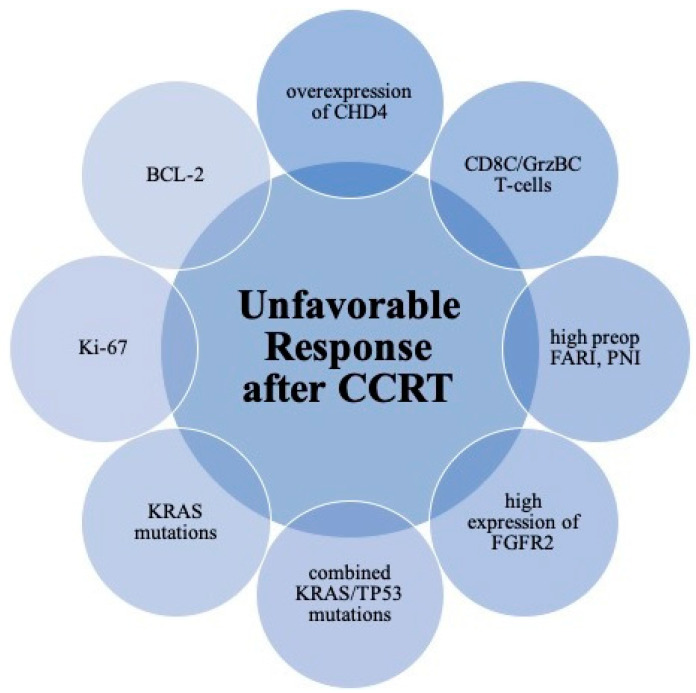
Schematic diagram of unfavorable biomarkers for neoadjuvant concurrent chemoradiation in patients with locally advanced rectal cancer. Abbreviations: CCRT, concurrent chemoradiation; preop, preoperative; FARI, Fibrinogen-Albumin ratio index; PNI, Prognostic Nutritional Index; FGFR, fibroblast growth factor receptor.

**Figure 3 cells-11-01611-f003:**
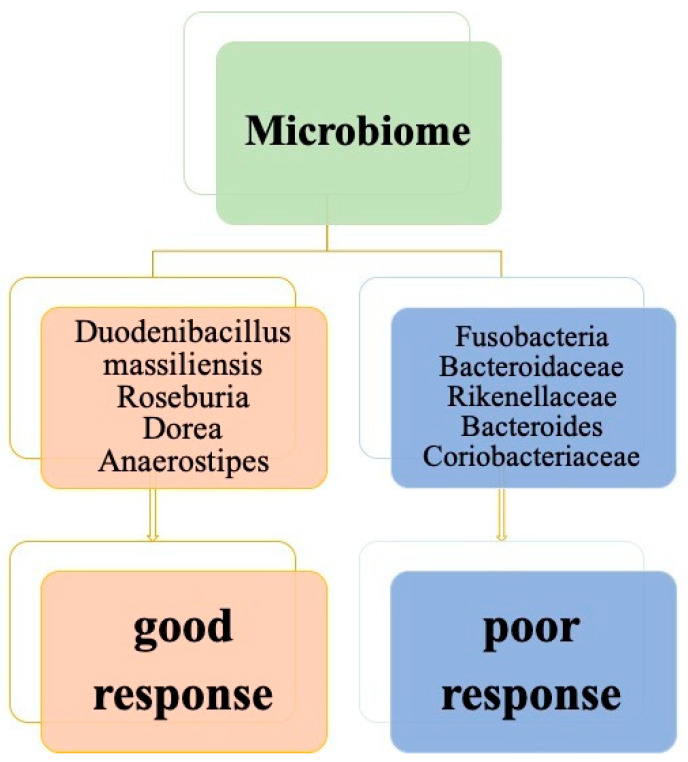
The different species of microbiota between favorable and unfavorable responses from neoadjuvant concurrent chemoradiation in patients with locally advanced rectal cancer.

**Table 1 cells-11-01611-t001:** Human studies of circulating tumor cells characteristics regarding tumor response after neoadjuvant treatment for locally advanced rectal cancer.

Author (Reference No.)	Study Year	CTC Type	Correlation with pCR or TRG	Patient Number
Tie et al. [8] * Prospective	2019	ctDNA	NO	*n* = 159
Sun et al. [25]	2013	400-/100-bp DNA ratio	YES	*n* = 103
Sun et al. [26]	2014	∆%CTC	YES	*n* = 34
Magni et al. [27] * Prospective	2014	∆CTC	YES	*n* = 85
McDuff et al. [28]	2021	ctDNA	YES	*n* = 29

Abbreviations: TRG: tumor regression grading; ctDNA: circulating tumor DNA; CTC: circulating tumor cell; pCR: pathological complete response; ∆%CTC: percentage difference in CTC counts between baseline and post-concurrent chemoradiation. * Prospective.

**Table 2 cells-11-01611-t002:** Molecular genetic markers: Oncogenes and tumor suppressors regarding tumor response after neoadjuvant treatment for locally advanced rectal cancer.

Author (Reference No.)	Study Year	Genes	Patient Number
Huang et al. [30]	2013	DPYD, TYMS, TYMP, TK1 and TK2	*n* = 60
Kim et al. [33]	2001	Ki-67	*n* = 23
Chen et al. [34]	2012	p53	*n* = 1830
Douglas et al. [36]	2020	ARID1A, PMS2, JAK1, CREBBP, MTOR, RB1, PRKAR1A, FBXW7, ATM C11orf65 and KMT2D	*n* = 17
Luna-Pérez et al. [45]	2000	KRAS	*n* = 37
Chow et al. [46]	2016	KRAS	*n* = 229
Davies et al. [47]	2011	Phospho-ERK, AKT	*n* = 70
Li et al. [50]	2019	KRAS, PDPK1, PPP2R5C, PPP2R1B and YES1	*n* = 6
Watanabe et al. [55]	2014	LRRIQ3, FRMD3, SAMD5 and TMC7	*n* = 52
He et al. [56]	2015	DNAJC12	*n* = 172
Huang et al. [57]	2015	SMAD3	*n* = 86
Huang et al. [58]	2020	ERCC	*n* = 20
Huang et al. [59]	2017	ERCC	*n* = 86
Kim et al. [60]	2007	95 genes	*n* = 46
He et al. [61]	2014	REG4	*n* = 172

## Data Availability

The data presented in this study are available from the corresponding author upon request.

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
