# Peer review of "Biomarkers of Favorable vs. Unfavorable Responses in Locally Advanced Rectal Cancer Patients Receiving Neoadjuvant Concurrent Chemoradiotherapy"

_cells, 2022, doi:10.3390/cells11101611_

Round 1

Reviewer 1 Report

This review artcle deals with the role of biomarkers in favorable or unfavorable response in locally advanced rectal cancer patients receiving neoadjuvant concurrent chemoradiotherapy.  The review describes in a descriptive manner the role of the old and the new biomarkers being used. The firures are also descriptive indicating in a humorous attitude  the favorable and unfavorable outcome. 

The abstract could be more indicative about the main subject referring to which groups of biomarkers is reffered.       

Author Response

Dear Reviewers and Editors of the Special Issue "The Research of Biomarkers in Colorectal Cancer and Gastric Cancer" of “Cells”,

  We are grateful for the opportunity to be reviewed by you. I am enclosing herewith a revised manuscript for the approval in terms of publication in the prestigious Cells. We have uploaded the improved version in accordance with your suggestions and attached the certificate for English editing managed by the language center of our University at the end of this cover letter.

  • To reviewer 1

We are appreciative of your recommendation. We have revised the abstract and indicated the groups of biomarkers as shown below. 

Colorectal cancer is the second cause of cancer death globally. The gold standard for locally advanced rectal cancer (LARC) is preoperative concurrent chemoradiation (CCRT) nowadays. Approximately three quarters of LARC patients do not achieve pathological complete response and hence suffer from relapse, metastases and inevitable death. The exploration of trustworthy and timely biomarkers for CCRT response is urgently called for. This review focused upon a broad spectrum of biomarkers including circulating tumor cells, DNA, RNA, oncogenes, tumor suppressor genes, epigenetics, impaired DNA mismatch repair , patient-derived xenograft, in vitro tumor organoid, immunity and microbiome. Utilizing proper biomarkers can assist in categorizing appropriate patients to the most efficient treatment modality with the best outcome accompanied by minimal side effects. The purpose of this review is to inspect and analyze accessible data in order to fully realize the promise of precision oncology for rectal cancer patients.

Reviewer 2 Report

This is a good review on biomarkers predicting the effect of chemoradiation in rectal cancer. 

I have some suggestions:

  1. Circulating DNA and miRNAs should be discussed separately, not under circulating tumour cells.
  2. There are some recent data on the dramatic effect of adding immunotherapy in MMR rectal cancer. Although this is not strictly chemotherapy, it should be mentionaed at least briefly, as MMR thus becomes a strong predictor of response to neoadjuvant therapy.
  3. Language revision is necessary. 

Author Response

Dear Reviewers and Editors of the Special Issue "The Research of Biomarkers in Colorectal Cancer and Gastric Cancer" of “Cells”,

  We are grateful for the opportunity to be reviewed by you. I am enclosing herewith a revised manuscript for the approval in terms of publication in the prestigious Cells. We have uploaded the improved version in accordance with your suggestions and attached the certificate for English editing managed by the language center of our University at the end of this cover letter. Please see the attachment for English certificate.

  •  
  • To reviewer 2

We appreciate your suggestions. We have separated the discussion regarding circulating DNA and miRNAs from circulating tumor cells according to your advice. The new paragraph (paragraph No. 3) is shown below. Immunotherapy for MMR rectal cancer is also revised in page 11-13, 9 new references were added. Language editing is provided by the language center of our Medical University as shown in the certificate.

Paragraph 3 DNA and RNA

However, Tie et al. reported that the conversion of circulating tumor DNA (ctDNA) status from positive at baseline to negative at four to six weeks after CCRT was not correlated with pCR (pCR vs non-pCR, 95% vs 88%, p = 0.46) in a prospective study [8], although a latest study involving 29 patients with LARC confirmed that ctDNA predicts pCR. The overall margin-negative, node-negative resection rate was 73% and was considerably higher among patients with undetectable preoperative ctDNA (n = 17, 88%) versus patients with measurable preoperative ctDNA (n = 9, 44%; p = 0.028) [28]. Table 1 summarized five LARC studies of CTCs and ctDNA characteristics regarding tumor response after neoadjuvant treatment.

Gene interaction network and module analysis of differential expression mRNAs contained in the lncRNA‐miRNA‐mRNA network identified five hub genes (KRAS, PDPK1, PPP2R5C, PPP2R1B, and YES1) closely associated with CCRT response in LARC [29]. Three lncRNA-based signatures: lnc-KLF7-1, lnc-MAB21L2-1, and LINC00324 were found to be the most promising variable subset for classification with overall sensitivity and specificity of 0.91 and 0.94 respectively, with an AUC of our ROC curve = 0.93 [30]. Palma et. al used Human WG CodeLink microarray platform and demonstrated that high Gng4, c-Myc, Pola1, and Rrm1 mRNA expression levels were significant predictors for CCRT response in LARC patients (p < 0.05) [31]. Aberrant DNA methylation, specifically the cytosine-phosphate-guanine (CpG) island methylator phenotype (CIMP), has been displayed in colorectal cancer [32]. Some researchers proposed a novel prophetic tool based on three CpGs potentially useful for pretreatment screening of LARC patients and steering the selection of treatment modality [33]. Lately, MicroRNA-130a (miR-130a) has been spectacularly upregulated in radiosensitive rectal cancer cells where overexpression of miR-130a promotes rectal cancer cell radiosensitivity by targeting SOX4 [34].

Paragraph 6, page 11-13

A recent study used immunohistochemistry, laser capture microdissection/qRT-PCR, flow cytometry, and functional analysis of tumor-infiltrating lymphocytes (TIL) from CRC patients and unveiled that the dynamic immune microenvironment of MSI was balanced by multiple counter-inhibitory checkpoints such as programmed cell death 1 (PD-1), PD-L1, cytotoxic T cell-associated protein 4 (CTLA-4), LAG-3, and IDO [72]. There have been good outcomes in stud-ies with immune checkpoint blockade by PD-1 receptor inhibitors, nivolumab and pembrolizumab, and CTLA-4 inhibitor, ipilimuma [73,74]. Avelumab is a fully human immuno-globulin that binds PD-L1. A phase II trial in patients with LARC, neoadjuvant radiation followed by mFOLFOX6 with avelumab is safe with a promising pCR [75].

For the minority patients with MSI-high/MMR deficient tumors, either due to Lynch syndrome or sporadic mutations, immunotherapy is recommended as first-line treatment [81]. There have been several case series of MMR deficient LARC that showed significant response with neoadjuvant immunotherapy-based systemic treatment [82-84]. Immune checkpoint inhibitors are more active in treatment-naïve patients than in those with refractory MSI-H/deficiency in MMR CRC [11,73,84]. Since neoadjuvant CCRT modulates immune-related characteristics of LARC, it may thus enhance the responsiveness of LARC to immunotherapy [85,86].
